# Type 1 Diabetes Mellitus, Psychopathology, Uncertainty and Alexithymia: A Clinical and Differential Exploratory Study

**DOI:** 10.3390/healthcare12020257

**Published:** 2024-01-19

**Authors:** Emanuele Maria Merlo, Rita Tutino, Liam Alexander MacKenzie Myles, Angela Alibrandi, Maria Carmela Lia, Domenico Minasi

**Affiliations:** 1Department of Biomedical and Dental Sciences and Morphofunctional Imaging, University of Messina, 98125 Messina, Italy; 2Pediatric Unit of Ospedali Riuniti Presidium, Grande Ospedale Metropolitano Bianchi Melacrino Morelli, 89124 Reggio Calabria, Italy; tutinorita11@gmail.com (R.T.); marilenalia@libero.it (M.C.L.); domenico.minasi@tiscali.it (D.M.); 3Department of Experimental Psychology, University of Oxford, Oxford OX1 2JD, UK; liam.a.myles@outlook.com; 4Department of Economics, University of Messina, 98122 Messina, Italy; aalibrandi@unime.it

**Keywords:** alexithymia, chronic disease, clinical psychology, intolerance of uncertainty, psychopathology, type 1 diabetes mellitus

## Abstract

Type 1 diabetes mellitus (T1DM) represents a complex pathology affecting a large number of people. Research suggests that psychological factors influence coping with T1DM. This study aimed to investigate the presence and role of psychopathology, alexithymia and uncertainty in people affected by T1DM. The sample consisted of 137 patients (88 females, 49 males) affected by T1DM aged from 11 to 19 years old (Mean: 13.87; SD: 2.40). The diagnostic protocol consisted of a sociodemographic questionnaire, Self-administration Psychiatric Scales for Children and Adolescents (SAFA), Toronto Alexithymia Scale-20 (TAS-20) and Intolerance to Uncertainty Scale-12 (IUS-12). Descriptive, differential, correlational and regression analyses were performed in order to examine the relationships between these variables. The results suggested the sample had high levels of psychopathological indexes, alexithymia and intolerance of uncertainty. Also, there were significant differences between TAS-20 and IUS-12 distributions with respect to psychopathology. Correlations and multivariate linear regressions indicated age, gender and education significantly predicted alexithymia and intolerance of uncertainty. This data suggest the presence of elevated psychopathology, alexithymia and uncertainty in people with diabetes.

## 1. Introduction

In line with various studies in the literature, chronic medical difficulties can cause psychological difficulties [1,2,3,4,5,6,7,8]. Indeed, the role of psychological variables has been studied with regard to the physical consequences of chronic medical difficulties, the role that psychopathology plays in the progression of chronic medical difficulties and finally, the management of physical therapies [9,10,11,12,13,14,15,16,17].

This research suggests that psychological facets exercise a central influence over the quality of life of patients with diabetes. In particular, Rubin and colleagues [9] studied the prevalence, manifestation, consequences, and treatment of psychological disorders in persons with diabetes. They found that dietary restrictions, self-monitoring of blood glucose, taking insulin injections, and lack of support from family and health care professionals predicted decreases in compliance and adherence to treatments as well as the development of psychological difficulties. Moreover, Atlantis and colleagues [10] and Talaei [13] highlighted that many patients suffering from psychopathology due to chronic diseases feel as though they are a burden.

Several studies in the literature have focused on variables such as alexithymia and intolerance of uncertainty, but there is a lack of knowledge regarding the relationship between emotional experience and pathology [18,19,20,21,22]. Authors such as Gibson and colleagues [19] have described the characteristics of intolerance of uncertainty in people suffering from diabetes, highlighting that prospective and inhibitory anxiety have an impact on these individuals through reduced quality of life and adverse biological factors. Marchini and colleagues [18] highlight the role of loss in the context of diabetes, studied both in dynamic and psychopathological terms, where loss produces mournful and depressive feelings in people with diabetes.

Barchetta and colleagues [22] highlight that diabetes can contribute to cognitive biases for past and present events, which can result in engagement in behaviors contrary to those indicated by medical professionals. As stated by Di Giuseppe and colleagues [23], the defensive patterns that patients can develop constitute a danger where the defense mechanisms cease to exercise a defensive role and take on psychopathogenic characteristics, as in the case of illness denial and psychosomatic manifestations of distress [24].

Alexithymia concerns the difficulty in describing, interpreting and identifying affective experiences. The role of alexithymia in chronic medical difficulties has often been emphasized in clinical research but must be considered within the context of the developmental stage of the medical condition [23,24,25,26,27,28,29,30,31,32]. The role of alexithymia in mental health difficulties is known in the literature, especially with reference to people experiencing chronic conditions [25,27]. Shang and colleagues [26] testify to the relationships between alexithymia and chronic conditions, identifying alexithymia as a significant predictor for the onset and maintenance of chronic conditions. The role of alexithymia and affective dysregulation is known in various healthcare settings and contexts, including the phenomenology related to diabetes.

Type 1 diabetes represents a condition with early onset and has the potential to give rise to psychological conditions that affect management of the difficulty and the quality of life of patients [33,34,35,36,37,38,39,40,41,42]. In particular, research suggests that management of chronic medical difficulties is influenced by alexithymia [43,44,45,46]. In a study published in 2014, Karukvi and Saarijärvi [43] identified the presence of alexithymia as a risk and prognostic factor for patient ill health. Particular reference must be made to the context of adolescence, where the ability to express issues related to affectivity is linked to the degree of development and psychoeducation of the individual [45]. Indeed, alexithymia’s role in diabetes and in the field of psychosomatic conditions has been well highlighted by previous studies [47,48,49,50,51,52,53]. As evidenced by studies in the literature, such as in the case of Taylor and colleagues [47], the physical implications of alexithymia on somatization of disorders are of fundamental importance. It is useful to note that not all populations affected by chronic conditions undergo psychodiagnostic evaluations useful for capturing adverse phenomena operating in a subclinical sense or fully active. Studies allow us to understand the phenomena on the basis of explanatory models that make it possible to use the concepts in the diagnostic-clinical and rehabilitation fields [49].

Despite self-management and empowerment policies, people with diabetes can transgress medical advice, which can produce organic damage [54,55,56,57,58]. From a clinical psychological point of view, it is important to understand what psychological facets contribute to transgressions of medical advice. Uncertainty may be a key variable since prospective and inhibitory anxiety due to a lack of knowledge can affect an individual’s mental health and behavior.

In line with current literature, the present study aimed to evaluate the presence of adverse factors in the quality of life of patients with T1DM. The variables taken into consideration included alexithymia, intolerance of uncertainty and the possible presence of psychopathology. The hypotheses of the study are presented below.

### Study Hypotheses

**Hypothesis** **1:***Psychopathology, alexithymia and intolerance of uncertainty are present in people with T1DM according to the normal and pathological group scores included in the validation studies and cut-offs*.

**Hypothesis** **2:***There are significant differences among the total scale scores of anxiety, depression, obsession, psychogenic eating disorders, somatic symptoms and hypochondria quartiles in people scoring highly on measures of alexithymia and uncertainty*.

**Hypothesis** **3:***There are significant differences among each of the provided psychopathology quartiles in people scoring highly on measures of alexithymia and intolerance of uncertainty (Hypothesis 2 is confirmed)*.

**Hypothesis** **4:***There are significant correlations among age, education, age of the diagnosis, alexithymia, difficulty in identifying and describing feelings, and eternally oriented thinking*.

**Hypothesis** **5:***Age, gender and education predict alexithymia and uncertainty*.

## 2. Materials and Methods

### 2.1. Participants and Procedures

The sample consisted of 137 patients with a female sex prevalence (Female: 88; Male: 46) aged from 11 to 19 years old (Mean: 13.87; SD: 2.40). The study was conducted from April 2023 to October 2023, all participants were affected by type 1 diabetes mellitus and were patients of the Pediatric Unit of the Ospedali Riuniti of Reggio Calabria. Recruitment was conducted during normal clinical activities of the Pediatric Unit, directed by Doctor Minasi. Inclusion criteria included age (10 to 20 years old), T1DM, and an absence of comorbidity. People presenting with other pathologies were excluded from the study. All patients were visited by an expert MD and involved in the study according to their consensus to participate. Participants were under pharmacological treatment for T1DM. Written informed consent was obtained from all participants and parents/tutors for minors. All participants were informed about the anonymous nature of the data and fully completed the protocol.

### 2.2. Ethics

All procedures were consistent with the 1964 Declaration of Helsinki and its later amendments or comparable ethical standards. The study was approved by the local Ethical Committee (Comitato Etico Regionale—Sezione Area Sud, Grande Ospedale Metropolitano “Bianchi-Melacrino-Morelli” of Reggio Calabria, No.: 19-2022, from 27 April 2022 onwards).

### 2.3. Instruments

The sociodemographic characteristics of the sample were obtained through a self-administration questionnaire examining age, gender, education (expressed in years) and age at diagnosis of type 1 diabetes mellitus.

#### 2.3.1. Intolerance of Uncertainty Scale-12 (IUS-12) Text

The Intolerance of Uncertainty Scale 12 (IUS-12) [59] is a 12-item scale dedicated to the clinical study of intolerance of uncertainty. Intolerance of uncertainty can be described as the tendency to react negatively to uncertainty regarding emotional, behavioral and cognitive feedback. IUS-12 is a self-report instrument using a 5-point Likert scale, ranging from “Not at all characteristic of me” to Entirely characteristic of me” (e.g., Item 1: *Unforeseen events upset me greatly*). The items are highly representative of the contents. The scale is composed of two main factors, prospective and inhibitory anxiety, and derives from a previous 27-item version of the scale known as IUS-27 [60,61]. The Italian version was adapted and validated through studies provided by Bottesi, Lauriola and colleagues [62,63]. According to Carleton and colleagues, who developed and validated the original version, the scale demonstrated consistent construct validity for total and subscale scores, internal reliability, and test–retest reliability (Cronbach’s α of 0.91, total scale, 0.85 for both subscale scores, r = 0.77) [59,64]. The Italian version reported high scores with reference to internal reliability, 0.80 for the IUS-12 total scale, 0.68 for prospective anxiety and 0.79 for inhibitory anxiety.

#### 2.3.2. Toronto Alexithymia Scale-20 (TAS-20)

The Toronto Alexithymia Scale (TAS-20) [65] is a 20-item self-report scale based on a 5-point Likert scale assessing alexithymia. Alexithymia can be described as the impossibility or severe difficulty in identifying and describing feelings and affective dynamics, followed by externally oriented thinking, which means a greater tendency to direct the thought to external dynamics rather than internal functioning. The 20 items are reported on a 5-point Likert scale ranging from Strongly Disagree to Strongly Agree (e.g., Item 1: “*I am often confused about what emotion I am feeling*”). The original version had a Cronbach’s alpha of 0.81 and its structure emerged as three main factors accounting for 31% of the total variance. TAS-20 represents a well-known and useful instrument to detect the presence of alexithymia in a wide range of groups. Regarding the three-factor structure, the main dimensions of the scale are difficulty identifying feelings (0.78), difficulty describing feelings (0.75) and externally oriented thinking (0.66). According to Bressi and colleagues (Italian Validation) [66], the cross-validation, including clinical and nonclinical samples, reported 0.77, 0.67 and 0.52 for the first, second and third factors, respectively; the scores of the clinical sample were 0.82 for the full scale, and 0.79, 0.68 and 0.54 for the three factors. Further studies have analyzed the psychometric properties of the scale, highlighting the good consistency and reliability of the three-factor structure.

#### 2.3.3. SAFA Scales

SAFA is a clinical instrument developed by Cianchetti and Sannio Fascello [67,68]. As a clinical psychometric test, it was validated in 2001. Its structure allows clinicians to complete a clinical investigation of the psychopathological conditions of tested participants. Its composition, despite being commonly presented as a unitary tool, is based on different scales assessing anxiety (SAFA Anxiety), depression (SAFA Depression), obsession (SAFA Obsession), somatic symptoms and hypochondria (SAFA Somatic symptoms and hypochondria), psychogenic eating disorders (SAFA Psychogenic eating disorders) and phobias (as nominal variables, not considered scale). Considering the used scales, SAFA Anxiety is composed of 50 items, SAFA Depression is composed of 50 items, SAFA Obsession is composed of 38 items, Safa Psychogenic eating disorders is composed of 30 items and SAFA Somatic symptoms and hypochondria of 25 items. All items are reported on a 3-point Likert Scale ranging from Not at all to Entirely. Referring to reliability, the original validation study considered both clinical and nonclinical participants. In these terms, the Cronbach’s alphas for SAFA Anxiety were 0.887 for the nonclinical sample and 0.956 for the clinical sample (test–retest Pearson *r*: 0.913, highly significant), 0.909 for the nonclinical sample and 0.943 for the clinical sample (test–retest Pearson *r*: 881, highly significant) of the SAFA Depression scale, 0.916 for the nonclinical sample and 0.895 for the clinical sample (test–retest Pearson *r*: 0.820) for the SAFA Obsession Scale, 0.814 for the nonclinical sample (test–retest Pearson *r*: 0.740, highly significant) for the SAFA Psychogenic eating disorder scale, and 0.876 for the nonclinical sample and 0.797 for the clinical sample (test-retest Pearson *r*: 0.567, highly significant) of the Somatic symptoms and hypochondria scale.

### 2.4. Statistical Analysis

Numerical data were expressed as means and standard deviations, and the categorical variables as numbers and percentages. The Kruskal–Wallis test was applied to assess statistically significant differences among quartiles of the SAFA Scales for the TAS-20 and IUS-12 scores. After the emergence of significant differences among quartiles, the Mann–Whitney test was adopted for the analysis of each couple. The Pearson correlation coefficient (*r*) test was used for the correlational analyses. Multivariate linear regressions were applied in order to highlight dependencies among a set of predictors (age, education and gender) and dependent variables referred to alexithymia and intolerance of uncertainty. A *p*-value smaller than 0.05 was considered to be statistically significant for multivariate linear regressions, and Kruskal–Wallis and Pearson correlation coefficient (*r*) tests. Using Bonferroni’s correction, a *p*-value smaller than 0.008 was considered to be significant for the Mann–Whitney test. Statistical analyses were performed using SPSS 26 for Windows package.

## 3. Results

### 3.1. Hypothesis 1

Descriptive statistics regarding Education, Age of the Diagnosis, SAFA, TAS-20 and IUS-12 variables are reported in Table 1.

Internal reliability coefficients (Cronbach alphas) for the used scale were 0.936 for SAFA Anxiety, 0.947 for SAFA Depression, 0.958 for SAFA Obsession, 0.934 for SAFA Psychogenic eating disorders, 0.953 for SAFA Somatic symptoms and hypochondria, 0.730 for TAS-20 total score, 0.792 for TAS-20 difficulty identifying feelings, 0.727 for TAS-20 difficulty describing feelings, 0.685 for TAS-20 externally oriented thinking, 0.765 for IUS-12 total score, 0.630 for IUS-12 prospective anxiety and 0.740 for IUS-12 inhibitory anxiety.

Considering the first hypothesis, the scores in the scales were, in most cases, over the normal scores included in the clinical validation studies and manuals. With reference to SAFA scales, the scores were higher than reference normal groups [67] for Safa Anxiety (study score: 51.255; reference mean normal group score: 25.44; pathological mean group score: 63.36), SAFA Depression (study score: 58.350; reference mean normal group score: 23.5; pathological mean group score: 69.89), SAFA Obsession (study score: 34.817; reference mean normal group score: 21.73; pathological mean group score: 56.44), SAFA Psychogenic eating disorders (study score: 27.963; reference mean normal group score: 15.17; pathological mean group score: 45.02) and SAFA Somatic symptoms and hypochondria (study score: 23.306; reference mean normal group score: 11.02; pathological mean group score: 24.43). Considering TAS-20 scores, the mean score was over the normal score cut-off (51) [66], demonstrating a borderline score for the whole sample and the presence of alexithymia over normal levels. With reference to the IUS-12 scores, in our sample, they were higher than in Italian validation studies [63] both for the IUS-12 total score (study score: 35.678; validation study score: 29.69), IUS-12 prospective anxiety (study score: 22.416; validation study score: 18.79) and IUS-12-inhibitory anxiety (study score: 13.622; validation study score 10.90).

### 3.2. Hypothesis 2

The Kruskal–Wallis test was used to identify statistically significant differences among SAFA quartiles compared to TAS-20 and IUS-12 factors. SAFA scale scores were divided into quartiles and considered with reference to each of the TAS-20 and IUS-12 factors. Table 2, Table 3, Table 4, Table 5 and Table 6 report data referred to the Kruskal-Wallis analyses.

Table 2 reports descriptive statistics for TAS-20 and IUS-12 factors and total scores, including the Kruskal–Wallis analysis for the SAFA Anxiety scale. There were significant differences among quartiles for the TAS-20 total score, difficulty identifying feelings, IUS-12 total score and inhibitory anxiety. This data showed it was possible to notice precise points of variation among factors’ quartiles of the SAFA Anxiety scale.

**Table 2 healthcare-12-00257-t002:** Descriptive statistics and Kruskal–Wallis analysis for SAFA Anxiety.

Descriptive Statistics								
SAFA Quartiles		TAS-20 Total Score	TAS-20 Difficulty Identifying Feelings	TAS-20 Difficulty Describing Feelings	TAS-20 Externally-Oriented Thinking	IUS-12 Total Score	IUS-12 Prospective Anxiety	IUS-12 Inhibitory Anxiety
**<Q1**	**Mean**	47.054	14.973	11.837	20.243	32.702	21.405	11.297
	**SD**	12.469	6.335	4.997	4.815	9.002	5.340	5.054
**Q1 ÷ Q2**	**Mean**	61.882	22.588	17.500	21.764	37.647	22.382	15.264
	**SD**	9.431	5.576	4.600	4.539	8.765	5.928	4.826
**Q2 ÷ Q3**	**Mean**	57.694	19.916	15.388	22.388	38.638	24.166	14.472
	**SD**	7.588	5.395	4.135	4.009	7.175	4.936	4.601
**>Q3**	**Mean**	48.133	15.133	12.266	12.266	33.566	33.566	11.966
	**SD**	10.702	7.025	5.362	4.315	9.216	4.745	5.423
**Total**	**Mean**	53.766	18.197	14.270	21.350	35.678	22.416	22.416
	**SD**	11.907	6.832	5.264	4.464	8.840	5.328	5.190
**Kruskal–Wallis**								
***p* value**		**<0.001**	**<0.001**	**<0.001**	0.304	**0.007**	0.117	**<0.001**
**Quartiles’ scores** **SAFA Anxiety**		Q1 = 36		Q2 = 51		Q3 = 69		

Significant *p* value < 0.05.

Table 3 reports the data emerged considering possible differences among TAS-20 and IUS-12 factors of SAFA Depression quartiles. Several significant differences emerged, showing significant differences for TAS-20 total score, difficulty identifying feelings, difficulty describing feelings, IUS-12 total score and inhibitory anxiety. No significant differences emerged with reference to externally oriented thinking and prospective anxiety.

**Table 3 healthcare-12-00257-t003:** Descriptive statistics and Kruskal–Wallis analysis for SAFA Depression.

Descriptive Statistics								
SAFA Quartiles		TAS-20 Total Score	TAS-20 Difficulty Identifying Feelings	TAS-20 Difficulty Describing Feelings	TAS-20 Externally-Oriented Thinking	IUS-12 Total Score	IUS-12 Prospective Anxiety	IUS-12 Inhibitory Anxiety
**<Q1**	**Mean**	48.647	15.705	12.441	20.500	35.147	22.147	13.000
	**SD**	11.308	6.969	5.472	4.272	9.963	5.662	5.892
**Q1 ÷ Q2**	**Mean**	58.975	21.317	16.439	21.219	37.463	23.146	14.317
	**SD**	11.862	5.824	5.000	5.012	8.267	5.067	4.317
**Q2 ÷ Q3**	**Mean**	57.000	20.321	15.607	21.964	37.678	23.357	14.321
	**SD**	10.150	5.888	4.263	3.995	7.448	5.478	5.106
**>Q3**	**Mean**	49.941	15.176	12.382	21.852	32.411	21.029	11.382
	**SD**	10.815	6.529	4.960	4.3632	8.711	5.078	4.905
**Total**	**Mean**	53.766	18.197	14.270	21.350	35.678	22.416	13.262
	**SD**	11.907	6.832	5.2643	4.464	8.840	5.328	5.190
**Kruskal–Wallis**								
***p* value**		**<0.001**	**<0.001**	**0.001**	0.718	**0.040**	0.212	**0.046**
**Quartiles’ scores** **SAFA Depression**		Q1 = 41.5		Q2 = 58		Q3 = 78		

Significant *p* value < 0.05.

**Table 4 healthcare-12-00257-t004:** Descriptive statistics and Kruskal–Wallis analysis for SAFA Obsession.

Descriptive Statistics								
SAFA Quartiles		TAS-20 Total Score	TAS-20 Difficulty Identifying Feelings	TAS-20 Difficulty Describing Feelings	TAS-20 Externally-Oriented Thinking	IUS-12 Total Score	IUS-12 Prospective Anxiety	IUS-12 Inhibitory Anxiety
**<Q1**	**Mean**	50.142	16.400	13.485	20.257	33.542	20.857	12.685
	**SD**	14.709	7.013	5.977	5.002	10.009	5.709	5.909
**Q1 ÷ Q2**	**Mean**	57.500	20.416	14.972	22.083	39.000	24.388	14.611
	**SD**	10.191	6.249	4.819	4.649	6.952	4.581	4.264
**Q2 ÷ Q3**	**Mean**	58.875	20.843	16.218	21.812	36.937	22.968	13.968
	**SD**	8.209	6.486	4.647	4.230	8.408	5.608	5.183
**>Q3**	**Mean**	48.735	15.205	12.500	21.264	33.176	21.411	11.764
	**SD**	10.366	5.998	4.937	3.816	8.733	4.843	5.039
**Total**	**Mean**	53.766	18.197	14.270	21.350	35.678	22.416	13.262
	**SD**	11.907	6.832	5.264	4.464	8.840	5.328	13.262
**Kruskal–Wallis**								
***p* value**		**<0.001**	**<0.001**	**0.020**	0.495	**0.008**	**0.022**	0.087
**Quartiles’ scores** **SAFA Obsession**		Q1 = 14		Q2 = 38		Q3 = 48.5		

Significant *p* value < 0.05.

Considering SAFA Obsession, several significant differences were found involving TAS-20 and IUS-12 factors. In particular, the significant differences concerned the TAS-20 total score, difficulty identifying feelings, difficulty describing feelings, IUS-12 total score and prospective anxiety. No significant differences emerged with reference to externally oriented thinking and inhibitory anxiety.

**Table 5 healthcare-12-00257-t005:** Descriptive statistics and Kruskal–Wallis analysis for SAFA Psychogenic eating disorders.

Descriptive Statistics								
SAFA Quartiles		TAS-20 Total Score	TAS-20 Difficulty Identifying Feelings	TAS-20 Difficulty Describing Feelings	TAS-20 Externally-Oriented Thinking	IUS-12 Total Score	IUS-12 Prospective Anxiety	IUS-12 Inhibitory Anxiety
**<Q1**	**Mean**	50.947	16.342	13.105	21.500	34.552	21.605	12.947
	**SD**	14.418	7.327	5.674	4.990	8.429	4.553	5.331
**Q1 ÷ Q2**	**Mean**	57.457	21.342	15.857	21.000	39.485	24.428	15.057
	**SD**	9.589	6.121	5.247	4.504	8.297	5.158	4.739
**Q2 ÷ Q3**	**Mean**	57.066	20.733	15.633	20.666	36.700	23.266	13.433
	**SD**	10.234	5.551	4.029	4.171	8.987	5.735	5.197
**>Q3**	**Mean**	50.205	14.794	12.735	22.147	32.117	20.500	11.617
	**SD**	10.859	5.855	5.206	4.098	8.343	5.287	5.093
**Total**	**Mean**	53.766	18.197	14.270	21.350	35.678	22.416	13.262
	**SD**	11.907	6.832	5.264	4.464	8.840	5.328	5.190
**Kruskal–Wallis**								
***p* value**		**0.011**	**<0.001**	**0.015**	0.478	**0.009**	**0.008**	**0.046**
**Quartiles’ scores** **SAFA Psychogenic eating disorders**		Q1 = 13		Q2 = 31		Q3 = 39.5		

Significant *p* value < 0.05.

Kruskal–Wallis test applied to SAFA Psychogenic eating disorders and considering possible differences among TAS-20 and IUS-12 factors’ quartiles, evidenced significant differences in the TAS-20 total score, difficulty identifying feelings, difficulty describing feelings, IUS-12 total score, prospective anxiety and inhibitory anxiety. In this case, the only variable not showing significant differences among quartiles was externally oriented thinking.

**Table 6 healthcare-12-00257-t006:** Descriptive statistics and Kruskal–Wallis analysis for SAFA Somatic symptoms and hypochondria.

Descriptive Statistics								
SAFA Quartiles		TAS-20 Total Score	TAS-20 Difficulty Identifying Feelings	TAS-20 Difficulty Describing Feelings	TAS-20 Externally-Oriented Thinking	IUS-12 Total Score	IUS-12 Prospective Anxiety	IUS-12 Inhibitory Anxiety
**<Q1**	**Mean**	48.628	15.342	12.685	20.600	34.257	21.657	12.600
	**SD**	12.840	6.919	5.449	4.525	9.274	5.390	5.4567
**Q1 ÷ Q2**	**Mean**	62.058	22.676	17.500	21.852	39.970	24.617	15.352
	**SD**	11.135	5.793	5.0527	5.046	7.432	4.722	4.715
**Q2 ÷ Q3**	**Mean**	55.764	19.676	14.500	21.588	35.764	22.000	13.764
	**SD**	7.647	5.103	4.287	4.271	9.557	5.851	5.393
**>Q3**	**Mean**	48.764	15.176	12.441	21.382	32.764	21.411	11.352
	**SD**	10.162	6.520	4.774	4.052	7.548	4.868	4.478
**Total**	**Mean**	53.766	18.197	14.270	21.350	35.678	22.416	13.262
	**SD**	11.907	6.832	5.264	4.464	8.840	5.328	5.190
**Kruskal–Wallis**								
***p* value**		**<0.001**	**<0.001**	**<0.001**	0.820	**0.003**	**0.015**	**0.012**
**Quartiles’ scores** **SAFA Somatic symptoms and hypochondria**		Q1 = 11		Q2 = 23		Q3 = 35.5		

Significant *p* value < 0.05.

Exploring differences referred to SAFA Somatic symptoms and hypochondria, TAS-20, total score, difficulty identifying feelings and difficulty describing feelings emerged as significantly different. In the IUS-12 case, all factors presented significant differences. In line with all analyses performed through the Kruskal–Wallis test, externally oriented thinking has never presented significant differences.

### 3.3. Hypothesis 3

According to the Hypothesis 3, the Mann–Whitney test was used to highlight the statistically significant differences among TAS-20 and IUS-12 quartiles that emerged through the Kruskal–Wallis test, as reported within Table 7, Table 8, Table 9, Table 10 and Table 11.

In the case of Anxiety, the Mann-Whitney test indicated several differences among TAS-12 and IUS-12 quartiles compared to the SAFA Anxiety scale. Starting with TAS-20 total score, significant differences were found between first and second, first and third, second and fourth, and third and fourth quartiles. The same results were obtained with reference to difficulty identifying feelings and difficulty describing feelings, highlighting the scale’s high consistency. Considering IUS-12 scales, the Kruskal–Wallis test showed significant differences only for IUS-12 total score and inhibitory anxiety. The only significant difference in the IUS-12 total score was between the first and third quartiles. In the case of inhibitory anxiety, significant values were found in the case of first and second, first and third, and second and fourth quartiles.

**Table 8 healthcare-12-00257-t008:** Differences among each factor of the TAS-20 and IUS-12 scales referring to SAFA Depression.

	Q1 vs. Q2	Q1 vs. Q3	Q1 vs. Q4	Q2 vs. Q3	Q2 vs. Q4	Q3 vs. Q4
**TAS-20 Total score**	**<0.001**	**0.001**	0.560	0.094	**<0.001**	0.037
**TAS-20 Difficulty identifying feelings**	**<0.001**	**0.004**	0.768	0.493	**<0.001**	**0.002**
**TAS-20 Difficulty describing feelings**	**0.002**	0.017	0.975	0.351	**0.001**	0.008
**IUS-12 Total Score**	0.259	0.263	0.246	0.826	0.011	0.013
**IUS-12 Inhibitory anxiety**	0.313	0.411	0.337	0.825	**0.007**	0.014
**Quartiles’ scores** **SAFA Depression**	Q1 = 41.5	Q2 = 58	Q3 = 78			

Significant *p*-value < 0.008 after Bonferroni’s correction. Bold values were significant values.

Considering Depression, several significant differences emerged among IUS-12 and TAS-20 quartiles. Starting with TAS-20 total score, significant differences emerged among the first and second, first and third, and second and fourth quartiles. The same results emerged considering difficulty identifying feelings, including the significant difference between the third and fourth quartiles. Significant differences involving difficulty describing feelings’ quartiles involved the first and second and second and fourth quartiles. In the case of IUS-12, the only significant differences that emerged were those between the second and fourth quartiles for both IUS-12 total score and inhibitory anxiety.

**Table 9 healthcare-12-00257-t009:** Differences among each factor of the TAS-20 and IUS-12 scales referring to SAFA Obsession.

	Q1 vs. Q2	Q1 vs. Q3	Q1 vs. Q4	Q2 vs. Q3	Q2 vs. Q4	Q3 vs. Q4
**TAS-20 Total score**	0.012	**0.003**	0.829	**0.002**	**0.002**	**<0.001**
**TAS-20 Difficulty identifying feelings**	0.009	0.008	0.560	**0.001**	**0.001**	**0.001**
**TAS-20 Difficulty describing feelings**	0.294	0.064	0.520	0.029	0.029	**0.003**
**IUS-12 Total Score**	0.012	0.106	0.805	**0.003**	**0.003**	0.030
**IUS-12 Prospective anxiety**	0.008	0.164	0.895	0.009	0.009	0.182
**Quartiles’ scores** **SAFA Obsession**	Q1 = 14	Q2 = 38	Q3 = 48.5			

Significant *p*-value < 0.008 after Bonferroni’s correction. Bold values were significant values.

Analyses considering obsession of TAS-20 and IUS-12 quartiles showed fewer significant values with regard to previous analyses. Significant differences in TAS-20 total score were among the first and third, second and third, second and fourth, and third and fourth quartiles. TAS-20 identifying feelings quartiles showed significant differences in the case of the second and third, second and fourth, and third and fourth quartiles. The only significant difference that emerged considering difficulty describing feelings was between the third and fourth quartiles. The IUS-12 total score showed significant differences emerged between the second and third and second and fourth quartiles. No significant differences emerged considering IUS-12 prospective anxiety quartiles.

**Table 10 healthcare-12-00257-t010:** Differences among each factor of the TAS-20 and IUS-12 scales referring to SAFA Psychogenic eating disorders.

	Q1 vs. Q2	Q1 vs. Q3	Q1 vs. Q4	Q2 vs. Q3	Q2 vs. Q4	Q3 vs. Q4
**TAS-20 Total score**	0.022	0.038	0.919	0.900	0.009	0.016
**TAS-20 Difficulty identifying feelings**	**0.003**	**0.004**	0.466	0.659	**<0.001**	**<0.001**
**TAS-20 Difficulty describing feelings**	0.028	0.039	0.799	0.668	0.015	0.016
**TAS-20 Externally-oriented thinking**	0.035	0.532	0.202	0.257	**0.001**	0.034
**IUS-12 Total Score**	0.015	0.312	0.173	0.387	**0.002**	0.036
**IUS-12 Prospective anxiety**	0.400	0.257	0.717	0.947	0.286	0.181
**IUS-12 Inhibitory anxiety**	0.124	0.809	0.283	0.198	0.004	0.122
**Quartiles’ scores** **SAFA Psychogenic eating disorders**	Q1 = 13	Q2 = 31	Q3 = 39.5			

Significant *p*-value < 0.008 after Bonferroni’s correction. Bold values were significant values.

Considering Eating Disorders, the statistically significant differences that emerged were regarding TAS-20 difficulty identifying feelings with reference to the first and second, first and third, second and fourth, and third and fourth quartiles, externally oriented thinking in the case of the second and fourth quartiles and IUS-12 total score showing a significant difference involving the second and fourth quartiles. No significant differences were found in the case of the other factors and scales.

**Table 11 healthcare-12-00257-t011:** Differences among each factor of the TAS-20 and IUS-12 scales referring to SAFA Somatic symptoms and hypochondria.

	Q1 vs. Q2	Q1 vs. Q3	Q1 vs. Q4	Q2 vs. Q3	Q2 vs. Q4	Q3 vs. Q4
**TAS-20 Total score**	**<0.001**	0.005	0.943	**0.004**	**<0.001**	0.009
**TAS-20 Difficulty identifying feelings**	**<0.001**	**0.002**	0.990	0.025	**<0.001**	**0.004**
**TAS-20 Difficulty describing feelings**	**<0.001**	0.147	0.942	0.011	**<0.001**	0.108
**IUS-12 Total Score**	**0.007**	0.417	0.490	0.061	**<0.001**	0.132
**IUS-12 Prospective anxiety**	0.008	0.718	0.732	0.044	**0.003**	0.132
**IUS-12 Inhibitory anxiety**	0.037	0.374	0.396	0.190	**0.001**	0.055
**Quartiles’ scores** **SAFA Somatic symptoms and hypochondria**	Q1 = 11	Q2 = 23	Q3 = 35.5			

Significant *p*-value < 0.008 after Bonferroni’s correction. Bold values were significant values.

SAFA Somatic symptoms and hypochondria concern somatic symptoms and hypochondria, so the emerged differences among TAS-20 and IUS-12 quartiles were regarding these fields. Most of the significant differences emerged between the first and second quartiles as well as the second and fourth quartiles. Surprisingly all values referred to the differences between the second and fourth quartiles were significant. Starting from the TAS-20 total score, differences among the first and second, second and third, and second and fourth quartiles were significant. Difficulty identifying feelings showed significant differences in the first and second, first and third, second and fourth, and third and fourth quartiles. Difficulty describing feelings showed two significant values regarding the first and second and second and fourth quartiles. All IUS-12 variables showed significant differences. IUS-12 total score highlighted significant values regarding the first and second and second and fourth quartiles. Prospective anxiety and inhibitory anxiety both showed significant differences between the second and fourth quartiles.

### 3.4. Hypothesis 4

Correlational analyses referred to Hypothesis 3 are reported in Table 12.

Correlational analysis showed several significant relationships. Starting from age, significant and positive correlations emerged involving TAS-20 total score, its externally oriented thinking factor and IUS-12 prospective anxiety. Education showed to be significant to age of the diagnosis and externally oriented thinking (TAS-20) and the emerged values were negative demonstrating opposite directions. Age of the diagnosis had a positive and significant correlation with Difficulties describing feelings (TAS-20). Considering these demographic variables, directions assumed by the above-mentioned significant correlations were positive for age and age of the diagnosis and negative for education. In this last case, education appeared to assume an opposite direction with reference to alexithymia. Several positive and significant correlations emerged for TAS-20 and IUS-12 total scores and related factors. Alexithymia showed the same direction assumed by intolerance of uncertainty, as in the case of the IUS-12 total score and prospective and inhibitory anxiety. The same significant and positive correlations were shown by both TAS-20 difficulty identifying and describing feelings of the IUS-12 total score and inhibitory anxiety. No significant correlations emerged for IUS-12 and TAS-20 externally oriented feelings.

### 3.5. Hypothesis 5

Multivariate linear regression analysis referred to Hypothesis 4 are reported in Table 13.

Multivariate linear regressions were used in order to highlight possible dependencies among the selected predictors. The TAS-20 and IUS-12 factors showed four significant values. Starting from age, a significant and positive dependency emerged with TAS-20 externally oriented thinking, showing how increasing age constitutes a causal predictor of improving this alexithymic factor. This datum appears to be in line with the literature considering externally oriented thinking regarding development. The second significant and positive dependency was with regard to gender (as a predictor) and TAS-20 difficulty identifying feelings. Considering the predictor’s values, this datum refers to female participants involved in the study. The third significant dependency to emerge was negative and involved education (predictor) and TAS-20 total score, highlighting how education assumes a positive role in diminishing alexithymia expression. The last one referred to education and externally oriented thinking, which was negative and in line with the previously stated, reinforcing the role of education in the decline of alexithymia.

## 4. Discussion

The results obtained through the analyses highlighted significant relationships between the variables. The first hypothesis concerned the presence of psychopathology, alexithymia and intolerance of uncertainty in diabetes. The results demonstrated the presence of psychopathology, alexithymia and intolerance of uncertainty in people with TWDM, consistent with the literature [10,13,14,69,70,71,72,73,74]. In particular, Popa-Varela and colleagues [11] highlighted the presence of psychopathology in people affected by T1DM. The study highlighted not only psychopathology but also the need for intervention to decrease pathological levels of anxiety, depression and somatic issues. Moreover, the authors defined this need as a challenge. Talaei [13] suggested the need to refer to adults, adolescents and children for support, with a view to understanding how the reality of people suffering from T1DM and T2DM is characterized by problems relating to mental health. Although in some studies the analyses stop at correlations, it is important to understand how some psychological variables play a causal role with respect to the chronic pathology and the quality of life of the participants. Şahin and colleagues [69] highlighted how issues relating to psychopathology not only impact those affected by the chronic disease but also how the issue extends to families and caregivers. In particular, the authors refer to the attitudes of parents and communities, emphasizing that awareness of such issues is necessary. The analyses of the studies cited, such as in the case of Skočić and colleagues [71] and Turin and Radobuljac [72], took into account the defense and coping phenomena of people affected by T1DM. As stated previously, peoples’ defensive patterns and coping methods can represent a strength. Likewise, the improper use of defenses and coping strategies can represent a risk factor. The use of coping and defenses can be protective, but it is equally known that strategies based on avoidance and improperly used defenses constitute an opportunity for the emergence of psychological problems. Any interventions must take into account not only the nature of these phenomena but also the fact that they essentially refer to developmental phases in which the participants’ cognitive functioning is not comparable to that of an adult. In terms of consequences, the study by Van Duinkerken and colleagues [75] highlights how clear the cognitive and psychological effects of living with a chronic pathology such as diabetes are, consistent with the results in this study.

The second hypothesis concerned the presence of significant differences between the scores related to psychopathology on measures of alexithymia and intolerance of uncertainty. Referring to this first step, we expected the emergence of significant differences among quartiles. The subsequent hypotheses highlighted precisely where the differences were present and significant. In all cases, significant differences were found in the psychopathological domain (e.g., Depression, Obsession, Somatic Symptoms; Table 2, Table 3, Table 4, Table 5, Table 6, Table 7, Table 8, Table 9, Table 10 and Table 11). Through subsequent differential analyses, it was possible to deduce the exact points of differentiation. Although it was possible to find clear indications about the relationships existing among depression, anxiety, obsessive thought, somatic symptoms and eating disorders [74,75,76,77,78,79,80,81,82,83,84,85], this approach to considering differences regarding alexithymia and intolerance of uncertainty in people with diabetes represented an innovative process. Indeed, it is possible to deduce precise distance thresholds in the distributions of the same scales. The detection of the presence of subthreshold symptoms represents an important starting point for the configuration of interventions since we are not faced with stable and evident psychopathological configurations on a symptomatic level [86]. In terms of primary prevention, testing for the presence of these subthreshold variables represents the first step. Unfortunately, in many cases, the study highlighted the presence of scores above the threshold, with a clear indication of how variables such as alexithymia and intolerance of uncertainty play a differential role in the expression of anxiety, depression, somatic symptoms, eating disorders, hypochondriasis and thought disorders. In other words, the emergence of psychopathological variables beyond the threshold calls for psychotherapeutic intervention and psychological support aimed at the conditions.

The third hypothesis concerned correlational analyses that included both demographic variables and those referring to alexithymia and uncertainty. The relationships that emerged predicted opposite directions between age and alexithymia, confirming a greater tendency to modulate affectivity corresponding to adulthood [57]. The same negative and significant direction was taken during the study years, confirming the constructive role of education on mental functioning [86,87,88,89,90,91]. With respect to the correlations between alexithymia and uncertainty, the data are consistent with previous studies. In fact, both on an epistemological and a clinical psychological level, the poor processing of the affective experience and the growing intolerance of uncertainty take the same maladaptive direction. Even more so, poor processing of emotions, feelings and mood dynamics would place the individual in the presence of both prospective and inhibitory anxiety, exactly as described by the factors implicated in the study of the scales. Although the presence of an increased risk of alexithymia, intolerance of uncertainty and psychopathology in people with T1DM is clear, and it is important to clarify how the issues manifest themselves and what relationships exist between the variables.

The last hypothesis took into consideration age, gender and education. The role of these predictors with respect to alexithymia and intolerance of uncertainty was statistically significant in four cases. Firstly, age appeared to be a significant predictor of externally oriented thinking, consistent with the literature. Gender significantly predicted difficulty in identifying feelings, with women finding it easier to identify feelings. The last two statistically significant findings involved education as a predictor of the total score and externally oriented thinking of the TAS scale. Both correlations were negative, suggesting that the role of education takes the opposite direction compared to alexithymia. These results appeared to be consistent with previous studies [45,85,86,87,88,92,93]. Nevertheless, further studies considering precisely participants affected by type 1 diabetes mellitus are needed. The role of the predictors highlighted by these analyses allows us to understand how variables such as age, education and gender influence equally important variables such as alexithymia and intolerance of uncertainty. In line with the literature, education represents a fundamental point of reference in the decrease of psychopathological phenomena and conditions adverse to the health of participants.

However, the study has limitations that need to be discussed. Being a cross-sectional study, it is not possible to determine causal relationships between the variables. Moreover, this study failed to examine the relationship between the aforementioned variables and biological variables related to T1DM. Future studies should consider biological variables and whether they influence participants’ biological status, compliance and adherence to treatments.

## 5. Conclusions

The present study considered clinically significant variables in a sample composed of those suffering from type 1 diabetes mellitus. The hypotheses of the study suggested the presence of consistent levels of psychopathology, alexithymia and intolerance of uncertainty. Beyond the descriptive statistics, several significant differences emerged in the scores on measures of alexithymia and intolerance of uncertainty, with respect to the psychopathological domains taken into consideration. It was possible to highlight a certain degree of suffering in the participants, often covered by variables of the order of alexithymia and expressed through anxiety and inhibition. Further studies should consider the statistical analyses performed in light of these hypotheses and the results in order to guide clinical decisions that are useful for helping people with diabetes and improving their quality of life.

## Figures and Tables

**Table 1 healthcare-12-00257-t001:** Descriptive statistics for clinical variables.

	Mean	SD	Min-Max (Scales Scores)	Min-Max (Study Scores)	Skewness	Kurtosis
**Education**	8.77	2.407	-	5–13	0.334	−1.003
**Age of the diagnosis**	7.91	3.394	-	0–16	−0.029	−0.719
**SAFA Anxiety**	51.255	20.916	0–100	5–96	−0.005	−0.850
**SAFA Depression**	58.350	24.027	0–112	9–101	−0.035	−0.819
**SAFA Obsession**	34.817	20.899	0–76	1–75	−0.029	−1.088
**SAFA Psychogenic eating disorders**	27.963	15.440	0–60	2–54	−0.110	−1.177
**SAFA Somatic symptoms and hypochondria**	23.306	14.913	0–50	0–50	−0.004	−0.164
**TAS-20 Total score**	53.766	11.907	20–100	22–82	−0.096	−0.362
**TAS-20 Difficulty identifying feelings**	18.197	6.832	7–35	7–33	0.097	−1.161
**TAS-20 Difficulty describing feelings**	14.270	5.264	5–25	5–25	0.032	−0.846
**TAS-20 Externally-oriented thinking**	21.350	4.464	8–40	8–30	−0.247	−0.353
**IUS-12 Total score**	35.678	8.840	12–60	15–55	0.086	−0.311
**IUS-12 Prospective anxiety**	22.416	5.328	6–30	8–34	−0.029	−0.151
**IUS-12 Inhibitory anxiety**	13.262	5.190	6–30	5–25	0.249	−0.843

**Table 7 healthcare-12-00257-t007:** Differences among each factor of the TAS-20 and IUS-12 scales referring to SAFA Anxiety.

	Q1 vs. Q2	Q1 vs. Q3	Q1 vs. Q4	Q2 vs. Q3	Q2 vs. Q4	Q3 vs. Q4
**TAS-20 Total score**	**<0.001**	**<0.001**	0.682	0.066	**<0.001**	**0.001**
**TAS-20 Difficulty identifying feelings**	**<0.001**	**<0.001**	0.850	0.047	**<0.001**	**0.004**
**TAS-20 Difficulty describing feelings**	**<0.001**	**0.002**	0.791	0.043	**<0.001**	**0.007**
**IUS-12 Total Score**	0.025	**0.003**	0.880	0.552	0.052	0.013
**IUS-12 Inhibitory anxiety**	**0.002**	0.009	0.531	0.433	0.010	0.048
**Quartiles’ scores** **SAFA Anxiety**	Q1 = 36	Q2 = 51	Q3 = 69			

Significant *p*-value < 0.008 after Bonferroni’s correction. Bold values were significant values.

**Table 12 healthcare-12-00257-t012:** The Pearson correlation coefficient (r) analysis.

	1.	2.	3.	4.	5.	6.	7.	8.	9.	10.
**1. Age**	-									
**2. Education**	**0.977 ****	-								
**3. Age of the diagnosis**	**0.216 ***	**0.189 ***	-							
**4. TAS-20 Total score**	**−0.178 ***	**−0.214 ***	0.087	-						
**5. TAS-20 Difficulty identifying feelings**	−0.037	−0.061	0.161	**0.824 ****	-					
**6. TAS-20 Difficulty describing feelings**	−0.009	−0.027	**0.175 ***	**0.811 ****	**0.648 ****	-				
**7. TAS-20 Externally-oriented thinking**	**−0.315 ****	**−0.354 ****	−0.127	**0.396 ****	−0.002	−0.051	-			
**8. IUS-12 Total Score**	0.078	0.051	−0.010	**0.446 ****	**0.450 ****	**0.360 ****	0.137	-		
**9. IUS-12 Prospective anxiety**	**0.177 ***	0.156	−0.026	**0.252 ****	**0.309 ****	0.155	0.046	**0.845 ****	-	
**10. IUS-12 Inhibitory anxiety**	−0.071	−0.073	0.044	**0.501 ****	**0.450 ****	**0.453 ****	0.094	**0.836 ****	**0.412 ****	-

Significant *p*-value: * <0.05; ** <0.001.

**Table 13 healthcare-12-00257-t013:** Multivariate linear regression analysis (Age, Gender and Education are selected predictors).

		TAS-20 Total Score	TAS-20 Difficulty Identifying Feelings	TAS-20 Difficulty Describing Feelings	TAS-20 Externally-Oriented Thinking	IUS-20 Total Score	IUS-12 Prospective Anxiety	IUS-12 Inhibitory Anxiety
**Age**	**B(CI: 95%)**	3.134 (−0.824/7.093)	914 (−1.393/3.221)	0.712 (−1.096/2.519)	1.471 (0.054/2.887)	2.365 (−0.654/5.384)	1.472 (−0.320/3.265)	0.893 (−0.877/2.662)
	** *p* **	0.120	0.435	0.438	**0.042**	0.124	0.107	0.320
	**VIF**	23.445	23.445	23.445	23.445	23.445	23.445	23.445
	**Tolerance**	0.043	0.043	0.043	0.043	0.043	0.043	0.043
**Gender**	**B(CI: 95%)**	1.653 (−2.614/5.921)	2.465 (−0.023/4.952)	0.583 (−1.366/2.532)	−0.875 (−2.402/0.652)	−0.278 (−3.533/2.977)	−1.261 (−3.194/0.672)	0.983 (−0.925/2.890)
	** *p* **	0.445	**0.042**	0.555	0.259	0.866	0.199	0.310
	**VIF**	1.089	1.089	1.089	1.089	1.089	1.089	1.089
	**Tolerance**	0.918	0.918	0.918	0.918	0.918	0.918	0.918
**Education**	**B(CI: 95%)**	−4.183 (−8.105/−0.261)	−1.156 (−3.442/1.130)	−0.776 (−2.567/1.015)	−2.061 (−3.464/−0.658)	−2.113 (−5.104/0.879)	−1.047 (−2.823/0.729)	−1.065 (−2.818/0.687)
	** *p* **	**0.037**	0.319	0.393	**0.004**	0.165	0.246	0.231
	**VIF**	23.025	23.025	23.025	23.025	23.025	23.025	23.025
	**Tolerance**	0.043	0.043	0.043	0.043	0.043	0.043	0.043
**F**		3.448	1.974	0.440	8.134	0.930	2.315	1.114
**Durbin-Watson**		2.088	1.905	2.037	1.801	1.694	1.690	1.768

Significant *p*-value < 0.05; Degree of Freedom: Regression DF = 3, Residual DF = 133, Total DF = 136 for all performed models.

## Data Availability

The data that support the findings of this study are available from the corresponding author, E.M.M., upon reasonable request.

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
