# Peer review of "Type 1 Diabetes Mellitus, Psychopathology, Uncertainty and Alexithymia: A Clinical and Differential Exploratory Study"

_healthcare, 2024, doi:10.3390/healthcare12020257_

Round 1
Reviewer 1 Report
Comments and Suggestions for Authors
This is a well-written and well-done paper aimed at exploring the levels and role of psychopathology, alexithymia, and uncertainty in a group of people affected by Type 1 Diabetes Mellitus (T1DM). The literature review is thorough, expressions are clear, and findings are significant and well explained. I have just some minor revisions to highlight.
1. Abstract: The authors should include the mean and standard deviation with reference to age and gender prevalence. The authors stated, " The sample consisted of 137 patients affected by T1DM aged from 11 17 to 19 years old.” Please provide more detailed information. Moreover, main operations should be stated in the methodological section to allow readers to grasp the methodological core of the research.
2. Keywords: I would suggest the authors include "Psychopathology."
3. Introduction: I would suggest the authors focus more on psychopathology associated with T1DM. Considering the study, this improvement would better support methods and related results. A short sub-paragraph including relevant studies would enhance the entire section. I appreciated the hypotheses.
4. Methods: Despite the authors presenting other data referring to Table 1, I would suggest including the mean and standard deviation even in paragraph 2.1. I appreciated the presentation of instruments. Regarding statistical analyses, it must be noted that this methodology is original. Despite a large number of studies including differential analyses, the analyses performed represent an innovative approach to understanding scales' internal differences related to psychopathological indexes.
5. Results: Tables and comments are clear. From line 138 to line 141, I suggest the authors expand this section, including a clear reference to hypotheses and their basis.
6. Discussion: I appreciated this section. I agree with the authors regarding a lack of knowledge referred to these differential analyses. Considering the exploratory nature of the study, it would represent a clear starting point useful to understand how alexithymia and intolerance to uncertainty distribution behave in the light of psychopathology.
My suggestion is to consider the article for acceptance after the above-mentioned minor revisions.
Author Response
Dear Reviewer, thank you for your comments. We believe they increased the value of the paper. Our responses are reported as follows.
This is a well-written and well-done paper aimed at exploring the levels and role of psychopathology, alexithymia, and uncertainty in a group of people affected by Type 1 Diabetes Mellitus (T1DM). The literature review is thorough, expressions are clear, and findings are significant and well explained. I have just some minor revisions to highlight.
Response: Dear Reviewer, thank you for appreciating our work. We believe it would represent a good clinical psychological article highlighting psychological and psychopathological difficulties experienced by subjects suffering from Type 1 Diabetes Mellitus.
- Abstract: The authors should include the mean and standard deviation with reference to age and gender prevalence. The authors stated, " The sample consisted of 137 patients affected by T1DM aged from 11 17 to 19 years old.” Please provide more detailed information. Moreover, main operations should be stated in the methodological section to allow readers to grasp the methodological core of the research.
Response: According to your comment we changed the section.
- Keywords: I would suggest the authors include "Psychopathology."
Response: We added the missing keyword.
- Introduction: I would suggest the authors focus more on psychopathology associated with T1DM. Considering the study, this improvement would better support methods and related results. A short sub-paragraph including relevant studies would enhance the entire section. I appreciated the hypotheses.
Response: Dear Reviewer, we agree with your suggestion. We enriched introduction following your indications.
- Methods: Despite the authors presenting other data referring to Table 1, I would suggest including the mean and standard deviation even in paragraph 2.1. I appreciated the presentation of instruments. Regarding statistical analyses, it must be noted that this methodology is original. Despite a large number of studies including differential analyses, the analyses performed represent an innovative approach to understanding scales' internal differences related to psychopathological indexes.
Response: Dear Reviewer, we agree with your comment. We believe that despite a large number of articles have been published on the themes we focused on, only a few items analysed differences in this way. In this sense, we count on the innovative style of methods.
- Results: Tables and comments are clear. From line 138 to line 141, I suggest the authors expand this section, including a clear reference to hypotheses and their basis.
Response: Done.
- Discussion: I appreciated this section. I agree with the authors regarding a lack of knowledge referred to these differential analyses. Considering the exploratory nature of the study, it would represent a clear starting point useful to understand how alexithymia and intolerance to uncertainty distribution behave in the light of psychopathology.
Response: Dear Reviewer, thank you for your suggestion. We believe that your comments contributed to improve the quality of the article.
My suggestion is to consider the article for acceptance after the above-mentioned minor revisions.
Reviewer 2 Report
Comments and Suggestions for Authors
Revie of the article: “Type 1 Diabetes Mellitus, Psychopathology, Uncertainty and 2 Alexithymia: A Clinical and Differential Exploratory Study”
The article considers important variables such as alexithymia, intolerance to uncertainty and psychopathology in subjects affected by type 1 diabetes mellitus. Being an exploratory study, this article constitutes relevant research in the field of chronic conditions and their psychological components. I think the article is well written, but it could be improved through minor changes.
In the abstract the authors reported “This study aimed to investigate the presence and role of psychopathology, alexithymia and uncertainty 16 in people affected by T1DM.” please make reference to differential analyses, correlations and dependences performed along the manuscript. More attention on hypotheses would improve the quality of the abstract.
Considering introduction, official definitions of alexithymia and intolerance to uncertainty would increase the value of the subsequent statements. I think the readers should benefit of high-quality definitions supported by relevant articles. In the light of psychopathology, the authors must also support each psychopathological domain with more studies. This section needs an implementation in order to reach a final form. After 1.1 Study hypotheses, before presenting the items the authors should include a brief introduction of the hypotheses. Considering hypotheses in general terms, I think that presenting the hypotheses along the manuscript before tables would lead the reader to immediately grasp the meaning of the results. Please suggest the hypotheses before each table.
I appreciated methods and results. Referring to this last section, authors must include the hypotheses reporting brief explanations in line with 1.1 section.
Finally, I think this article has potential and could represent a good contribution considering Journal’s aim and scopes.
Author Response
Dear Reviewer, thank you for appreciating the article and for suggesting the required minor changes. According to your suggestions, we provided all required changes. Our responses are reported as follows.
The article considers important variables such as alexithymia, intolerance to uncertainty and psychopathology in subjects affected by type 1 diabetes mellitus. Being an exploratory study, this article constitutes relevant research in the field of chronic conditions and their psychological components. I think the article is well written, but it could be improved through minor changes.
Response: Dear Reviewer, thank you for your positive feedback. According to your comments, we performed all required changes.
In the abstract the authors reported “This study aimed to investigate the presence and role of psychopathology, alexithymia and uncertainty 16 in people affected by T1DM.” please make reference to differential analyses, correlations and dependences performed along the manuscript. More attention on hypotheses would improve the quality of the abstract.
Response: Dear Reviewer, we performed the changes according to your comment. Missing information has been added.
Considering introduction, official definitions of alexithymia and intolerance to uncertainty would increase the value of the subsequent statements. I think the readers should benefit of high-quality definitions supported by relevant articles. In the light of psychopathology, the authors must also support each psychopathological domain with more studies. This section needs an implementation in order to reach a final form. After 1.1 Study hypotheses, before presenting the items the authors should include a brief introduction of the hypotheses. Considering hypotheses in general terms, I think that presenting the hypotheses along the manuscript before tables would lead the reader to immediately grasp the meaning of the results. Please suggest the hypotheses before each table.
Response: Dear Reviewer, we implemented the highlighted sections in order to follow your comments.
I appreciated methods and results. Referring to this last section, authors must include the hypotheses reporting brief explanations in line with 1.1 section.
Response: Dear Reviewer, as you can see, we performed the required changes.
Finally, I think this article has potential and could represent a good contribution considering Journal’s aim and scopes.
Reviewer 3 Report
Comments and Suggestions for Authors
1. The introduction and hypotheses seem unspecific. Provide a wider theoretical context, highlighting previous reports on the study variables, and based on this please indicate the relevance of your study.
2. Line 69, 91: Typos here and in many other places.
3. Line 73: Indicate clearly the number of females and males.
4. When the study was conducted? Inclusion/exclusion criteria? Mean age? Other demographic variables?
5. Taking into account the clinical character of the study, the participants section is poor. Presence of comorbidity, course of disease, etc.
6. Zeros before full stops in number were used not acc. to the journal's guidelines.
7. The measures were described somewhat illogically. Please step by step describe measures. What do they measure? How many items? How many factors? Examples of statements? Response scales? And after that indicate psychometric properties of Italian versions.
8. IUS-12: Italian version?
9. Hypotheses should not be described using variables names (e.g., "Correlational relationships among age, years of study, age of the diagnosis, TAS-20 and IUS-12 variables"). Use psychological language to indicate hypothesis.
10. SAFA scales were described extremely poorly. Please reconsider.
11. "The Spearman test" what is it? Please use adequate statistical terminology. the paper must be reconsidered.
12. Statistical section consists of repeatings the same sentences. Please reconsider.
13. Table 1: Years of study? What is it? Education?
14. Table 1: Please indicate skewness, kurtosis, min and max range, your range, as well as internal reliability coefficitnets obtiained in your study.
15. Lines 138-141: This description is unclear. What did you did here?
16. SAFA quartiles: Please indicate M and SD.
17. In "post hoc" tables clearly indicate what group has lower or higher results. For example, create an additional column and indicate this. Now, the readability of the paper is somewhat poor.
18. You have to many abbreviations regarding SAFA scales. You write for people, thus, SAFA A etc. you can replace by SAFA Anxiety etc.
19. Table 12: *Bold values were significant values (*<0.005; ** <0.001). Are you sure that * p < 0.005? Not 0.05?
Moreover, this is sub-standard statistical description "Bold values were significant values". Clearly indicate what you mean here > p-values.
In general, as I mentioned above, statistical analysis has a lot of problems with presentation and descriptions. Please consult with a statistician how to present this clearly.
20. Indicate clearly a typ of correlation analysis.
21. Table 13: Extremely sub-standard. Please describe clearly where your predictors, independent variables... And in general, please see high-quality papers how to present regression analyses.
Indicate clearly VIF, Tolerance, Durbin-Watson statistics... CI: indicate clearly 95%? 90%?....
22. Limitations of the study?
23. Practical implications?
24. The discussion section should be expanded. It is somewhat poor.
25. Future directions?
The paper should be very extensively rewritten. In the current form, the paper can not be considered for publication as there are many unclear issues, especially regarding the methodology and statistical analyses.
Comments on the Quality of English LanguageStatistical descriptions are sub-standard.
Author Response
Dear Reviewer, thank you for your comments. We believe they improved the quality of the article. All authors contributed to the revisions and with particular reference to methodological, statistical and linguistic fields, AA Professor of Medical Statistics and LAMM Clinical Psychologist, English Native Speaker working in a prestigious British University, fully revised the above-mentioned aspects in order to reach clarity and scientific accuracy. As you can see along the new version of the manuscript, the changes derived from your comments are highlighted in yellow. Thank you again for your contribution.
- The introduction and hypotheses seem unspecific. Provide a wider theoretical context, highlighting previous reports on the study variables, and based on this please indicate the relevance of your study.
Response: Dear Reviewer, thank you for your comment. According to your suggestions we enriched introduction and hypothesis with greater resonance for the included studies. We believe that this path would increase the value of the study.
- Line 69, 91: Typos here and in many other places.
Response: Dear Reviewer, thank you for your comment. As you can see the whole manuscript has been reviewed by LAMM, colleague and English Native Speaker. Thank you for noticing these issues.
- Line 73: Indicate clearly the number of females and males.
Response: Dear Reviewer, we provided for missing information
“The sample consisted of 137 patients with a female sex prevalence (Female: 88; Male: 88) aged from 11 to 19 years old (Mean: 13.87; SD: 2.40).”
- When the study was conducted? Inclusion/exclusion criteria? Mean age? Other demographic variables?
Response: Dear Reviewer, thank you for your comments. We provided for missing information.
“The sample consisted of 137 patients with a female sex prevalence (Female: 88; Male: 88) aged from 11 to 19 years old (Mean: 13.87; SD: 2.40). The study was conducted from April 2023 to October 2023, all subjects were affected by Type 1 Diabetes Mellitus and patients of the Pediatric Unit of the Ospedali Riuniti of Reggio Calabria. Recruitment was conducted during normal clinical activities of the Pediatric Unit, directed by Doctor Minasi. Inclusion criteria were referred to age, T1DM, absence of comorbidity. Subjects presenting other pathologies were excluded from the study.”
- Taking into account the clinical character of the study, the participants section is poor. Presence of comorbidity, course of disease, etc.
Response: Dear reviewer, according to the emerged need we rephrased the section.
“The sample consisted of 137 patients with a female sex prevalence (Female: 88; Male: 88) aged from 11 to 19 years old (Mean: 13.87; SD: 2.40). The study was conduct-ed from April 2023 to October 2023, all subjects were affected by Type 1 Diabetes Mellitus and patients of the Pediatric Unit of the Ospedali Riuniti of Reggio Calabria. Recruitment was conducted during normal clinical activities of the Pediatric Unit, di-rected by Doctor Minasi. Inclusion criteria were referred to age, T1DM, absence of comorbidity. Subjects presenting other pathologies were excluded from the study. All patients were visited by an expert MD and involved in the study according to their consensus to participate. Subjects were under pharmacological treatment with refer-ence to T1DM. Written informed consent was obtained from all participants and par-ents/tutors for minors. All participants were informed about the anonymous nature of data and fully completed the protocol.”
- Zeros before full stops in number were used not acc. to the journal's guidelines.
Response: Thank you, we solved the issue.
- The measures were described somewhat illogically. Please step by step describe measures. What do they measure? How many items? How many factors? Examples of statements? Response scales? And after that indicate psychometric properties of Italian versions.
Response: Dear reviewer, according to your comments we rephrased the instrument sections according to the suggested structure:
-Name of the scale
-Number of items
-Definition of treated phenomena
-Items examples
-Ordinal scale included in the instruments
-Factors
-Indexes emerged in the validation studies
-Studies referred to linguistic adaptation and related indexes (if needed)
This structure has been provided for all involved instrument.
“The Intolerance of Uncertainty Scale 12 (IUS-12) [59] is 12-item scale dedicated to the clinical study of intolerance to uncertainty. Intolerance to uncertainty can be described as the tendency to react negatively to uncertainty in the light of emotional, behavioral and cognitive feedbacks. IUS-12 is a self-report instrument based on a 5-point Likert scale. The items are highly representative of the contents and followed by a 5-point Likert scale ranging from “Not at all characteristic of me” to Entirely characteristic of me” (e.g., Item 1: Unforeseen events upset me greatly). The scale is composed by two main factors, Prospective and Inhibitory anxiety, and derives from a previous 27-item scale version known as IUS-27 [60,61]. The Italian version adapted and validated through studies provided by Bottesi, Lauriola and colleagues [62,63]. According to Carleton and colleagues who developed and validated the original version, the scale demonstrated consistent construct validity for total and subscale scores, internal reliability, and test-retest reliability (Cronbach's α of .91, total scale, .85 for both subscale scores, r = .77) [59,64]. The Italian version reported high scores with reference to internal reliability, 0.80 for the IUS-12 total scale, 0.68 for prospective anxiety and 0.79 for inhibitory anxiety.”
“The Toronto Alexithymia Scale (TAS-20) [65] is a 20-item self-report scale based on a 5-point Likert scale assessing alexithymia. Alexithymia can be described as the impossibility or the severe difficulty in identifying, describing feelings and affective dynamics, followed by externally oriented thinking meant as a greater tendency to direct the thought on external dynamics rather than internal functioning. The 20 items are followed by a 5-point Likert scale ranging from Strongly Disagree to Strongly Agree (e.g., Item 1: “I am often confused about what emotion I am feeling”). The original version had a Cronbach’s alpha of .81 and its structure emerged as three main factors accounting the 31% of the total variance. TAS-20 represents a well-known and useful instrument to detect the presence of alexithymia in a wide range of groups. Regarding to the three-factor structure, the main dimensions of the scale are: difficulty identifying feelings (.78), difficulty de-writing feelings (.75) and externally-oriented thinking (.66). According to Bressi and colleagues (Italian Validation) [66], the cross validation including clinical and non-clinical samples reported .77, .67 and .52, respectively, for the first, second and third factors; the scores of the clinical sample were .82 for the full scale, .79, .68 and .54 for the three factors. Further studies have analyzed the psychometric properties of the scale, highlighting the good consistency and reliability of the three-factor structure.”
“Safa is a clinical instrument developed by Cianchetti and Sannio Fascello [67,68]. As a clinical psychometric test, it was validated in 2001. According to his structure, it allows clinicians to complete a clinical investigation of the psychopathological conditions of tested subjects. Its composition, despite being commonly presented as a unitary tool, is based on different scales assessing anxiety (SAFA Anxiety), depression (SAFA Depression), obsession (SAFA Obsession), somatic symptoms and hypochondria (SAFA Somatic symptoms and hypochondria), Psychogenic eating disorders (SAFA Psichogenic eating disorders) and phobias (as nominal variables, not considered scale). Considering the used scales, SAFA Anxiety is composed by 50 items, SAFA Depression is composed by 50 items, SAFA Obsession is composed by 38 items, Safa Psychogenic eating disorders is composed by 30 items and SAFA S by 25 items. All items are followed by a 3-point Likert Scale ranging from Not at all to Entirely. Referring to reliability, the original validation study considered both clinical and non-clinical subjects. In these terms the Cronbach’s alphas for SAFA Anxiety were .887 for non-clinical sample and .956 for the clinical sample (test-retest Pearson r: .913, highly significant), .909 for non-clinical sample and .943 for the clinical sample (test-retest Pearson r: 881, highly significant) of SAFA Depression scale, .916 for non-clinical sample and .895 for the clinical sample (test-retest Pearson r: .820) for SAFA Obsession Scale, .814 for the non-clinical sample (test-retest Pearson r:.740, highly significant) for SAFA Psychogenic eating disorders scale, .876 for non-clinical sample and .797 for the clinical sample (test-retest Pearson r: .567, highly significant) of Somatic symptoms and hypochondria scale.”
- IUS-12: Italian version?
Response: Thank you for asking about an implementation of these sections. According to your comments we improved the specific paragraphs including missing information.
- Hypotheses should not be described using variables names (e.g., "Correlational relationships among age, years of study, age of the diagnosis, TAS-20 and IUS-12 variables"). Use psychological language to indicate hypothesis.
Response: Done.
Hypothesis 1: Presence of psychopathological components, alexithymia and intolerance to uncertainty in people with T1DM;
Hypothesis 2: Differences among anxiety, depression, obsession, psychogenic eating disorders, somatic symptoms and hypochondria quartiles in the light of alexithymia and uncertainty;
Hypothesis 3: Differences among each of the provided psychopathological quartiles in the light of alexithymia and intolerance to uncertainty (Hypothesis 2 is confirmed);
Hypotheses 4: Correlational relationships among age, education, age of the diagnosis, alexithymia, difficulty in identifying and describing feelings, eternally oriented thinking;
Hypothesis 5: Age, gender and education will predict alexithymia and uncertainty.
- SAFA scales were described extremely poorly. Please reconsider.
Response: Dear Reviewer, thank you for your request. Done.
- "The Spearman test" what is it? Please use adequate statistical terminology. the paper must be reconsidered.
Response: Dear reviewer, the Spearman Test was used for the reasons foreseen in the statistical paragraph:
“The non-parametric approach was used since non-normality was verified for most of the variables examined. In order to evaluate the correlations among the variables the Spearman's rank correlation coefficient test was used.”
We rephrased the section in order to improve the readability of the section.
- Statistical section consists of repeatings the same sentences. Please reconsider.
Response: Dear reviewer, thank you for your suggestion. As you can see, the section has been rephrased.
“Numerical data were expressed as means and standard deviations, and the categorical variables as numbers and percentages. The non-parametric approach was used since non-normality was verified for most of the variables examined. In order to evaluate the correlations among the variables the Spearman's rank correlation coefficient test was used. Kruskal-Wallis test applied to assess statistically significant differences among quartiles of the TAS-20 and IUS-12 scales in the light of clinical SAFA Scales. After the emergence of significant differences among quartiles, the Mann-Whitney test was adopted for the analysis of each couple. A p-value smaller than .05 was considered to be statistically significant for Spearman and Kruskal-Wallis tests. Considering the Bonferroni’s correction, a p-value smaller than 0.008 was considered to be significant for the Mann-Whitney test. Statistical analyses were performed using SPSS 26 for Windows package.”
- Table 1: Years of study? What is it? Education?
Response: Dear Reviewer, thank you for your comment. We agree, “Education” is the right word. As you can see along the manuscript, “years of study” was replaced with “Education”.
- Table 1: Please indicate skewness, kurtosis, min and max range, your range, as well as internal reliability coefficitnets obtiained in your study.
Response: Dear Reviewer, thank you for your request. We believe it improves the value of the section. As suggested, we provided for Min-Max scores for each variable (validation studies and instruments), Min-Max scores for the study variables, Skewness and Kurtosis. Moreover, according to your comment we provided for internal reliability coefficients obtained in the study.
- Lines 138-141: This description is unclear. What did you did here?
Response: Dear Reviewer, sorry for this section. It contained some errors. The new paragraph including the right form replaced the one you mentioned:
“The Kruskal Wallis test was used to identify statistically significant differences among SAFA quartiles compared to TAS-20 and IUS-12 factors. SAFA scales scores were divided into quartiles and considered with reference to each of the TAS-20 and IUS-12 factors.”
- SAFA quartiles: Please indicate M and SD.
Response: Dear reviewer, SAFA quartiles M and SD are reported in tables 2, 3, 4, 5 and 6 (highlighted data).
- In "post hoc" tables clearly indicate what group has lower or higher results. For example, create an additional column and indicate this. Now, the readability of the paper is somewhat poor.
Response: Dear reviewer, thank you for your comment. We are sorry if the data did not appear immediately evident to you, according to the previous response, we have placed the reader in the position of interpreting on the basis of the descriptive statistics (therefore tables 2, 3, 4, 5 and 6), including required indexes so that it is possible to evaluate greater or lower scores relating to the quartiles, such as to highlight where the higher score declines to the significant difference. According to our Medical Statistics Professor involved in the study (AA), please mind tables 2, 3, 4, 5 and 6 for Mean and Standard Deviation.
- You have to many abbreviations regarding SAFA scales. You write for people, thus, SAFA A etc. you can replace by SAFA Anxiety etc.
Response: We replaced abbreviations with full names. Please note, we previously used abbreviation since the original study named the variables exactly how we used them. However, we appreciated and understood your comment, so that we followed your indication.
- Table 12: *Bold values were significant values (*<0.005; ** <0.001). Are you sure that * p < 0.005? Not 0.05?
Response: Dear Reviewer, thank you for the comment. It was an error to indicate “0.005” instead of “0.05”. It has been corrected.
Moreover, this is sub-standard statistical description "Bold values were significant values". Clearly indicate what you mean here > p-values.
Response: Corrected.
In general, as I mentioned above, statistical analysis has a lot of problems with presentation and descriptions. Please consult with a statistician how to present this clearly.
Response: Dear Reviewer, thank for your suggestion. AA, one of the authors is a Professor of Medical Statistics. We are now confident about statistics.
- Indicate clearly a typ of correlation analysis.
Response: Dear Reviewer, thank you. We provided for:
“Table 12. Spearman's rank correlation coefficient analysis.”
- Table 13: Extremely sub-standard. Please describe clearly where your predictors, independent variables... And in general, please see high-quality papers how to present regression analyses.
Indicate clearly VIF, Tolerance, Durbin-Watson statistics... CI: indicate clearly 95%? 90%?....
Response: Dear Reviewer, thank for your comment. As you can see according to table 13, new columns including VIF, Tolerance and DURBIN-Watson statistics were provided. As you can note, being a matter of collinearity, VIF and Tollerance produced the same indexes for each column (TAS-20 and IUS-12 variables). Data was represented according to your indication and we provided for CI rate, as highlighted in yellow. We agree with the need to represent Dubrin-Watson statistics. Now you can find all required indexes within a dedicated column.
- Limitations of the study?
Response: Dear Reviewer, we included more reference to study limitations in the conclusions. Thank for your comment.
- Practical implications?
Response: According to your suggestion, more reference to practical implication was included.
- The discussion section should be expanded. It is somewhat poor.
Response: Dear Reviewer, according to your comment we enriched and rephrased the discussion in order to reach a better form.
- Future directions?
Response: Future directions were included in the conclusions. Thank you for your comment.
The paper should be very extensively rewritten.
Response: Most of the sections have been rewritten according to your comments. Thank you for your help.
In the current form, the paper can not be considered for publication as there are many unclear issues, especially regarding the methodology and statistical analyses.
Response: According to your comments, the whole work has been revised. In particular, all authors contributed to the changes. With specific reference to methodology and statistical analyses, one of the authors (AA), Professor of Medical Statistics, fully reviewed the above-mentioned sections in order to perform the required changes. We are now confident about methodology and analyses. With reference to linguistic aspects, the paper has been reviewed in depth by one of the authors (LAMM) who is an English Native Speaker working in a prestigious British University. According to the performed changes, we hope you will find the current form acceptable.
Thank you again for your help revising the manuscript and suggesting important changes.
Round 2
Reviewer 3 Report
Comments and Suggestions for Authors
Dear Authors,
Thanks for your efforts in reconsidering your paper. Unfortunately, these efforts were insufficient to improve the paper. The paper has serious theoretical and methodological problems as well as statistical problems and errors.
1. Inconsistencies with zeros before full stops in numbers were not addressed adequately. This is somewhat disappointing, considering the fact that the first author of the paper is a Guest Editor of this research topic.
2. Hypotheses were presented unspecifically and in some cases incorrectly, e.g., "Hypothesis 1: Presence of psychopathological components, alexithymia and intolerance to uncertainty in people with T1DM". All people has some levels of alexithymia and intolerance to uncertainty. How did you check the presence of alexithymia? Based on what cut-off scores? This was not presented in your paper. See lines 201-207: where did you refer to your hypothesis 1? There is no information. Therefore, the paper does not meet rigorous scientific criteria.
3. The issue with SAFA P and SAFA S still remains (see previous comments).
4. Lines 206-207: Internal consistency reliability coefficients should be presented for separate subscales (scores) and total scores.
5. Mean and SD in some tables should be presented before Min-Max (Scores).
6. Table 2 and tables before: SAFA quartiles. Indicate scores of these quartiles clearly.
7. p-values of < 0.001 should be presented as < 0.001, not p = 0.000. In general, this is incorrect, as p-values can not equal 0. Please see statistical literature on this issue. This (*p value <.05.) in the notes for tables is misleading, as you note p-values of p < 0.001 as being p < 0.05. Use common statistical descriptions with * p < 0.05, ** p < 0.01, and *** p < 0.001. In your cases, you report exact values, there is no need to report * here.
8. Lines 193-194: "The non-parametric approach was used since non-normality was verified for most of the variables examined". In your case, you can use parametric tests as skewness and kurtosis values are between -1 and 1 in most cases, and your sample.
9. Table 13: Table 13 and related analyses are sub-standard. The presentation of regression analysis in very uncommon. In general, predictors are presented in the left side, and dependent variables are presented at the top of the tables.
It is fully unclear how many regressions models were conducted. Are there 21 models (7 dependent variables with predictor age plus 7 dependent variables with predictor gender and 7 dependent variables with predictor education), or 7 models with 7 dependent variables and three predictors (i.e., age, gender and education) within each model.
10. Basic regression parameters, i.e., F, residuals and df for each regression model, were not presented. The authors did not describe whether they used dummy variables for nominal variables. Moreover, abbreviations, e.g., TAS-20 F1 or F2 or F3 are not described.
General comments:
As mentioned above, the statistical analysis section is poor. The correctness of these analyses seems uncertain.
Moreover, the idea of the study is unclear. In some cases, the authors focus on SAFA, then on sociodemographic variables, and some only alexithymia and uncertainty. In general, it seems that the authors want to find and present something statistically significant.
Despite the authors' efforts in improving the paper, the paper has serious theoretical, methodological and statistical problems and errors.
Author Response
Dear Reviewer, below are our responses to your comments and suggestions.
Thanks for your efforts in reconsidering your paper. Unfortunately, these efforts were insufficient to improve the paper. The paper has serious theoretical and methodological problems as well as statistical problems and errors.
Response: Dear Reviewer, thank you for evaluating our paper once again. According to your suggestions we hope the paper will reach a publishable form.
- Inconsistencies with zeros before full stops in numbers were not addressed adequately. This is somewhat disappointing, considering the fact that the first author of the paper is a Guest Editor of this research topic.
Response: Dear Reviewer, as suggested the inconsistences were corrected.
- Hypotheses were presented unspecifically and in some cases incorrectly, e.g., "Hypothesis 1: Presence of psychopathological components, alexithymia and intolerance to uncertainty in people with T1DM". All people has some levels of alexithymia and intolerance to uncertainty. How did you check the presence of alexithymia? Based on what cut-off scores? This was not presented in your paper. See lines 201-207: where did you refer to your hypothesis 1? There is no information. Therefore, the paper does not meet rigorous scientific criteria.
Response: Dear Reviewer, hypotheses were rephrased according to your suggestions and the hypothesis one was followed by necessary explanations. Moreover, after hypothesis 1 we reported a new paragraph including the requested comparisons with normal and pathological groups of validation studies.
- The issue with SAFA P and SAFA S still remains (see previous comments).
Response: Dear Reviewer, the issue has been finally solved.
- Lines 206-207: Internal consistency reliability coefficients should be presented for separate subscales (scores) and total scores.
Response: Dear Reviewer, according to your suggestion we provided for internal consistency coefficients for both scales and related factors.
- Mean and SD in some tables should be presented before Min-Max (Scores).
Response: Dear reviewer, according to your suggestion, mean scores are now represented before Min-Max scores.
- Table 2 and tables before: SAFA quartiles. Indicate scores of these quartiles clearly.
Response: Dear Reviewer, SAFA quartiles were reported for Tables 2, 3, 4, 5, 6, 7, 8, 9, 10, 11. Thank you for your suggestion.
- p-values of < 0.001 should be presented as < 0.001, not p = 0.000. In general, this is incorrect, as p-values can not equal 0. Please see statistical literature on this issue. This (*p value <.05.) in the notes for tables is misleading, as you note p-values of p < 0.001 as being p < 0.05. Use common statistical descriptions with * p < 0.05, ** p < 0.01, and *** p < 0.001. In your cases, you report exact values, there is no need to report * here.
Response: Dear Reviewer, according to your suggestions we corrected the values.
- Lines 193-194: "The non-parametric approach was used since non-normality was verified for most of the variables examined". In your case, you can use parametric tests as skewness and kurtosis values are between -1 and 1 in most cases, and your sample.
Response: Dear reviewer, we corrected the statistical analysis section and replaced the table reporting the new analyses performed through the Pearson correlation coefficient (r) test.
- Table 13: Table 13 and related analyses are sub-standard. The presentation of regression analysis in very uncommon. In general, predictors are presented in the left side, and dependent variables are presented at the top of the tables. It is fully unclear how many regressions models were conducted. Are there 21 models (7 dependent variables with predictor age plus 7 dependent variables with predictor gender and 7 dependent variables with predictor education), or 7 models with 7 dependent variables and three predictors (i.e., age, gender and education) within each model.
Response: Dear Reviewer, according to your suggestion we followed the instructions. Now predictors are on the left and dependent variables are on the first row. Following your instructions we replaced the previous table with the one you can find. The new one follows the order you suggested.
- Basic regression parameters, i.e., F, residuals and df for each regression model, were not presented. The authors did not describe whether they used dummy variables for nominal variables. Moreover, abbreviations, e.g., TAS-20 F1 or F2 or F3 are not described.
Response: Dear Reviewer, according to your suggestions we reported the missing parameters. Thank you for your help. Moreover, we did not use dummy variables because they were not foreseen by the models we performed.
General comments:
As mentioned above, the statistical analysis section is poor. The correctness of these analyses seems uncertain.
Moreover, the idea of the study is unclear. In some cases, the authors focus on SAFA, then on sociodemographic variables, and some only alexithymia and uncertainty. In general, it seems that the authors want to find and present something statistically significant.
Despite the authors' efforts in improving the paper, the paper has serious theoretical, methodological and statistical problems and errors.
Response: Dear Reviewer, we believe that following your precise instructions the paper reached a better form. Our efforts precisely considered your suggestions in order to solve the emerged issues. Thank you for your efforts in the review of this contribution.
